Accepted at the ICLR 2024 Workshop on AI4Differential Equations In Science

# Estimating Field Parameters from Multiphysics Governing Equations with Scarce Data

**Xuyang Li, Mahdi Masmoudi, Nizar Lajnef & Vishnu Naresh Boddeti**
Department of Civil Engineering and Computer Science Engineering
Michigan State University
East Lansing, MI 48824, USA
`{lixuyan1,masmoud2,lajnefni,vishnu}@msu.edu`

## Abstract

Real-world physical phenomena often involve complex, coupled behaviors influenced by spatially distributed physical properties. This complexity poses significant challenges for modeling, particularly when faced with limited or noisy observations. In this work, we introduce NeuroPIPE for field parameters inference in partial differential equations (PDEs). We employ deep neural networks to model the field parameters while solving the PDEs in discretized form using the finite difference method (FDM). Focusing on a representative example of cardiac electrophysiology with just 400 measurements, NeuroPIPE accurately captures the underlying parameters and physical behaviors. We demonstrated that our approach surpasses the state-of-the-art physics-informed neural networks (PINN) in terms of model robustness, parameter estimation accuracy, and training efficiency. Even with abundant training data, PINN fails in parameter inference in some cases, whereas NeuroPIPE consistently performs well. Additionally, NeuroPIPE achieves significantly higher inference accuracy by one order of magnitude. Our approach holds substantial promise for learning and understanding many complex physics problems with a significant reduction of training data.

## 1 Introduction

Over the past few years, deep learning powered by physics knowledge has transformed how complex physical phenomena are learned. However, capturing and modeling those phenomena governed by differential equations remains a complex and time-consuming task. This complexity usually arises from the non-constant nature of governing parameters in PDEs and the frequent occurrence of multi-domain physical processes, which current data-driven methods may struggle to address.

Recently, physics-informed machine learning (ML) methods such as using PINN (Chen et al., 2021), sparse regression (Brunton et al., 2016), Fourier neural operators (Li et al., 2020), and Neural-FEM (Pan & Duraisamy, 2020) have demonstrated an exceptional capability to model various physical phenomenon. These methods allow the direct learning of physical behaviors or discover the partially known governing differential equations based on observed data. However, they are primarily constrained to forward inference of physical phenomena rather than inverse parameter estimation.

Besides, modern ML methods and their applications pose several challenges in real-world physics problems. *First*, many parameter estimation methods are data-driven and failed to consider physics constraint. These approaches may have limitations in real-world applicability, particularly in scenarios that involve predicting future events and adapting to changes in boundary and initial conditions. *Second*, the lack of field parameters assumptions within highly nonlinear systems (Chen et al., 2021), particularly in multiphysics or coupled problems, can lead to significant discrepancies in modeling engineering responses. *Third*, the scarcity of real-world measurements poses a significant challenge for many ML methods, which require substantial amounts of training data to accurately estimate parameters or capture the underlying physics (Li et al., 2022).

In this article, we tackle the aforementioned challenges by introducing NeuroPIPE (see Figure 1), a parameter estimation framework employing neural networks and FDM. NeuroPIPE operates under the assumption that observed physics is represented by nonlinear parametric PDEs, where the unknown parameter fields characterize the physical phenomena in the computational domain. To address this, we initialize deep neural networks to model the unknown parameters in accordance with their dependency. Subsequently, the PDE system is discretized using FDM and solved with a differential equations solver. During neural network training, the spatial-temporal predictions are compared to the measured data, and the resultant errors are minimized. Following that, the optimized neural network model can accurately estimate the target parameters and effectively predict the physics behavior even beyond the training region (extrapolation).

In summary, our contributions to ML and differential equations are as follows:

- We introduce a framework named NeuroPIPE utilizing neural networks to estimate parameters that characterize various PDEs directly.
- NeuroPIPE can model various coupled problems. We employed a cardiac electrophysiology problem and demonstrated accurate dynamics modeling with temporal extrapolation.
- We compare NeuroPIPE with the PINN approach by highlighting its robustness in estimating diverse unknown field distributions, superior precision in parameter estimation, faster convergence, and its ability to operate effectively under sparse and noisy training data.

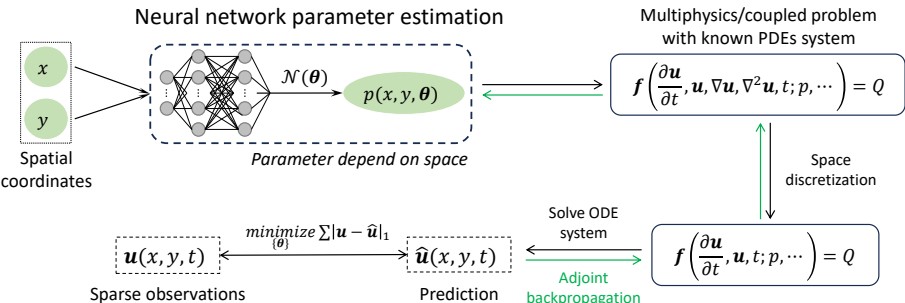

Figure 1: **Overview**. The physical phenomena are often characterized by parametric PDEs. Deep neural networks (referred to as $\mathcal{N}$) can effectively model parameter distribution. By minimizing the error between model predictions and observations, neural networks are optimized to provide accurate parameter estimation and forward response modeling.

## 2 RELATED WORK

Parameter estimation has been extensively studied in single-physics problems assuming scalar parameters. Various methods are employed, including finite element updating (Steenackers & Guillaume, 2006; Ebrahimian et al., 2017), Bayesian neural networks (Yang et al., 2021), least squares estimation (Ji et al., 2020), Kalman filter (Varshney et al., 2019; Hossain et al., 2022), Gaussian process (Zhang & Gu, 2022; Deng et al., 2020), and sparse identification (Brunton et al., 2016; Chen et al., 2021). However, in approaches like Bayesian methods, limitations arise from the assumption that unknown parameter values adhere to a prior distribution, which might be impractical when dealing with unknown field variables. And sparse identification is often limited to scalar parameters (Chen et al., 2021). In addition, the considerable non-linearity in the system poses difficulties for inverse estimation when employing statistical approaches for parameter estimation.

Recently, PINN has been known for integrating domain knowledge into neural network models and modeling various physics phenomena efficiently, addressing forward problems (Cai et al., 2021; Bolandi et al., 2023) and inverse problems such as parameter estimation (Herrero Martin et al., 2022; Zhao et al., 2022). Particularly, PINNs can model field parameters in scenarios involving complex multiphysics or coupling effects (Tartakovsky et al., 2020; Taneja et al., 2022), and the field parameters and state variables are often learned and predicted simultaneously with separate neural networks. However, achieving optimal performance with PINN still demands a considerable amount of training samples (Li et al., 2022; He et al., 2022).

## 3 METHOD

The proposed NeuroPIPE discretizes PDEs spatially using FDM and utilizes differential equation solvers for inference (see Figure 1). The spatial derivatives within PDEs are approximated by central difference (Randall, 2005). Next, the field parameter $p$ is modeled by a feed-forward neural network with spatial coordinates inputs $x$ and $y$. The inputs are scaled from -1 to 1 for better training performance. The network in this study consists of 6 layers, with each of the middle four layers containing 150 neurons and featuring skip-connections. The network parameters are denoted as $\boldsymbol{\theta}$.

$$p = \mathcal{N}(x, y, \boldsymbol{\theta}) \tag{1}$$

The estimated parameter from the neural network is inserted into differential equations for forward inference (state variable prediction). The inference is compared with available observations in mini-batches to compute the loss. The adjoint sensitivity method (Chen et al., 2018; Rackauckas et al., 2019) is utilized for efficient backpropagation and network training.

## 4 RESULTS

To validate NeuroPIPE's superior performance in field parameter estimation, we utilize a cardiac electrophysiology application (Zaman et al., 2021; Ntagiantas et al., 2024). Research efforts are dedicated to estimating heterogeneous cardiac tissue electrical conductivity using silico data, particularly for detecting fibrosis associated with arrhythmias or atrial fibrillation (Ntagiantas et al., 2024). The coupled model takes the form of a reaction-diffusion system (Herrero Martin et al., 2022).

$$\frac{\partial V}{\partial t} = \nabla(D\nabla V) - k_0 V(V - a)(V - 1) - VW \tag{2a}$$

$$\frac{\partial W}{\partial t} = \left(\epsilon + \frac{\mu_1 W}{V + \mu_2}\right)\left(-W - k_0 V(V - b - 1)\right) \tag{2b}$$

where the diffusion tensor $D$ determines the propagation speed and is proportional to the electrical conductivity of the tissue $\sigma$. $a$ and $b$ are known scalar related to the tissue excitation threshold and refractoriness (Herrero Martin et al., 2022). The state variable $V$ represents the membrane voltage and is accessible in experimental measurements. $W$ denotes an unknown state variable. $k_0 = 8, \mu_1 = 0.2, \mu_2 = 0.3, a = 0.01, b = 0.15$, and $\epsilon = 0.002$.

We employ a 2D slab of cardiac tissue ($1cm \times 1cm$) as the spatial domain and aim to recover heterogeneous diffusion tensor $D(x, y)$. Healthy tissue is represented by $D = 0.1mm^2/TU$, while fibrosis tissue is represented by $D = 0.02mm^2/TU$. $1TU$ is roughly $13ms$ (Herrero Martin et al., 2022). Neumann boundary condition is applied with $\frac{\partial V}{\partial x} = 0$ and $\frac{\partial V}{\partial y} = 0$.

During training, Gaussian noise is added to the training data (observed $V$) to mimic the real-world situations (see Figure 2a). The training data covers the first 40 $TU$ at intervals of 1 $TU$, matching the timestep used for solving the discretized PDE. In Figure 2b-c, we thoroughly examine the challenges posed by training data scarcity by illustrating parameter estimation errors for $D$ and forward inference errors for $V$, respectively. As tested, the number of observations is ultimately reduced to 800 while still accurately capturing the parameter distribution variance. Overall, NeuroPIPE demonstrates robust and accurate field parameter estimation performance, even in scenarios involving noisy data and a limited number of training points (i.e., the parameter distributions can refer to the second row of Figure 2d).

In addition, NeuroPIPE are compared with a PINN (Herrero Martin et al., 2022). Examining the predicted physical response of $V$ at a random spatial point in Figure 2a, NeuroPIPE strictly adheres to the physics, predicting non-negative $V$ values and accurately matching the initial condition compared to PINN. Furthermore, NeuroPIPE showcases robustness in making precise temporal extrapolations beyond the training region, after $40TU$. For parameter estimation comparisons, we conducted two cases, illustrated in Figure 2d. NeuroPIPE can successfully recover parameters in both cases with only 400 data, while PINN fails when estimating the random distribution $D$ in case 2 even with abundant data. Subsequently, utilizing case 2, the forward $V$ predictions are compared in Figure 2e. In the first row, PINN achieves good predictions within the training region but still has

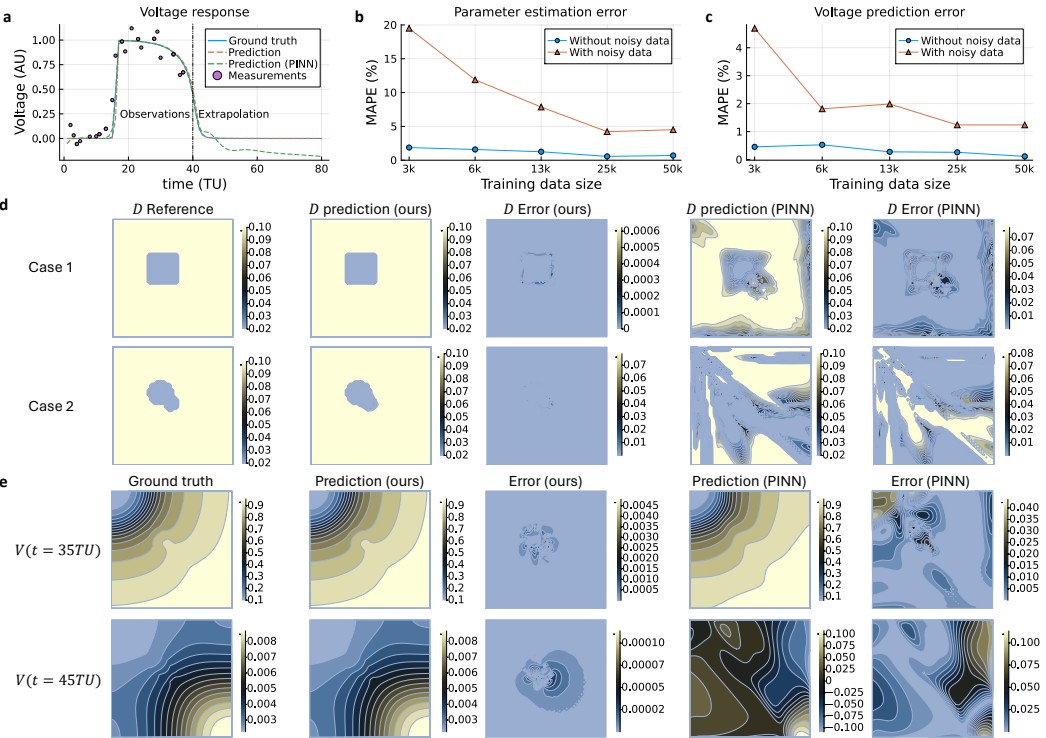

Figure 2: Parameter estimation for a cardiac electrophysiology application. NeuroPIPE outperforms PINN in all accuracy comparisons (parameter estimation, forward inference including interpolation and extrapolation). **a**. The training region with Gaussian noise, and the forward inference performance comparisons between NeuroPIPE and PINN. **b**. The parameter estimation error under scarce data. **c**. The forward inference error under scarce data. **d**. Parameter estimation results for two different cases. **e**. The forward inference compression between NeuroPIPE and PINN in case 1.

one order magnitude higher error than NeuroPIPE. Additionally, PINN exhibits much higher errors for temporal extrapolation (the second row of Figure 2e) due to inaccurate estimation of parameter $D$. On the other hand, while we only demonstrate cases with discrete distributions, it is important to note that neural network-based NeuroPIPE are capable of recovering various field distributions.

Moreover, training a PINN demands significantly higher computational resources and memory. For parameter estimation in case 1, PINN necessitates over 200,000 iterations of training on an NVIDIA RTX A6000 GPU (approximately 40 times slower if using a CPU). NeuroPIPE on the other hand, only necessitates 4,000 iterations on the same CPU, and the total training time remains half as long.

## 5  CONCLUSION

In this paper, we presented NeuroPIPE for parameter field estimation from governing PDEs with scarce data. We extensively explored its application in an emerging multiphysics field where the target parameter is unknown but crucial for characterizing physics phenomena. Our findings demonstrate the high robustness of NeuroPIPE across different parameter distributions. The estimated parameter closely aligns with the reference, even using 400 training data. In comparison, PINN struggles with estimating complex parameter distributions, exhibiting much lower estimation accuracy and less accurate forward inference. NeuroPIPE is also expected to be highly efficient when handling scalar parameters or single physics problems. Future applications could extend to other PDEs such as diffusion equations, Burgers equations, heat transfer, and advection problems.

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
