# OpenReview forum: "Estimating Field Parameters from Multiphysics Governing Equations with Scarce Data"
_ICLR.cc/2024/Workshop/AI4DiffEqtnsInSci — AI4DiffEqtnsInSci @ ICLR 2024 Poster_

### Official Review · Reviewer_eqk5 · 2024-02-26
**Review of “Estimating Field Parameters from Multi-Physics Governing Equations with Scarce Data”.**

**Rating:** 7
**Confidence:** 4

**Review:**

This work proposes a new method for data-driven parameter estimation. They use a neural network to estimate parameters and finite difference methods to compute derivatives and integrate the system forward in time for computing a loss. Parameter estimation is an important problem, particularly in the context of learning from experimental data. Several existing methods have been proposed for these types of problems, but all have different benefits and drawbacks. The novelty of this paper is to compute the loss with respect to the solution of the ODEs discretized using finite difference methods. The authors compare the performance of this method against PINNs on a cardiac electrophysiology dataset where they estimate the value of the diffusion tensor. The authors also explore the impact of the amount of training data on the performance of the model and show that it performs well, even with noise.
The method introduced in the paper is interesting and the results are promising, but the paper could use more details on what methods (both machine learning and finite difference) are implemented in the work and explore additional test cases, even very simple ones.

Primary Concerns
1)	The paper includes no details on the specific architecture used or the time stepping used to solve the discretized ODEs.
2)	The method is applied to learn a single derivative in a set of ODEs, which contain no other derivatives. Applying this same method to other problems that have more complex expressions may lead to the accumulation of error using finite difference to evaluate the system and thus poor convergence. This may affect performance in certain systems referenced in the conclusion, such as the Burgers equations.
3)	The parameter fields estimated in this setup are quite discrete, I would be interested to see if the model struggles in estimating fields that vary continuously across the domain. This may simplify the task by removing sharp edges, but also may make the impact of estimating incorrectly less apparent during the integration.
4)	In this instance it appears that the cost of integrating the model is significant relative to computing and updating gradients. (My rough estimate is that the model trains 50x less epochs than PINN but trains for half the time, so each training step is 25 times more expensive.) This cost may further increase in systems that demand smaller time steps, require more complex numerics, or may decorrelate more slowly and thus require longer integrations. This is a worthwhile tradeoff in the example shown, but it is important to acknowledge that this cost may not always scale favorably.

---

### Meta-Review · Area_Chair_hM4t · 2024-03-01

**Recommendation:** Accept (Poster)

**Metareview:**

This paper presents NeuralFD for estimating field parameters within PDE. which is able to captures the physics behaviors and the underlying parameter distributions. The concerns of clarification raised by the reviewers are valid. I strongly command the author to address those concerns in the camera ready version.

---

### Decision · Program_Chairs · 2024-03-01

Accept (Poster)